# Career intentions of medical students in the UK: a national, cross-sectional study (AIMS study)

Tomas Ferreira ![ORCID] ,[1] Alexander M Collins,[2] Oliver Feng,[3] Richard J Samworth,[3] Rita Horvath,[1] The AIMS Collaborative

[1]School of Clinical Medicine, University of Cambridge, Cambridge, UK
[2]School of Public Health, Faculty of Medicine, Imperial College London, London, UK
[3]Statistical Laboratory, Centre for Mathematical Sciences, University of Cambridge, Cambridge, UK

**Correspondence to**
Mr Tomas Ferreira;
tf385@cam.ac.uk

## ABSTRACT

**Objective** To determine current UK medical students' career intentions after graduation and on completing the Foundation Programme (FP), and to ascertain the motivations behind these intentions.

**Design** Cross-sectional, mixed-methods survey of UK medical students, using a non-random sampling method.

**Setting** All 44 UK medical schools recognised by the General Medical Council.

**Participants** All UK medical students were eligible to participate. The study sample consisted of 10 486 participants, approximately 25.50% of the medical student population.

**Outcome measures** Career intentions of medical students postgraduation and post-FP, motivations behind these career intentions, characterising the medical student population and correlating demographic factors and propensity to leave the National Health Service (NHS).

**Results** The majority of participating students (8806/10 486, 83.98%) planned to complete both years of the FP after graduation, with under half of these students (4294/8806, 48.76%) intending to pursue specialty training thereafter. A subanalysis of career intentions after the FP by year of study revealed a significant decrease in students' intentions to enter specialty training as they advanced through medical school. Approximately a third of surveyed students (3392/10 486, 32.35%) intended to emigrate to practise medicine, with 42.57% (n=1444) of those students not planning to return. In total, 2.89% of students intended to leave medicine altogether (n=303). Remuneration, work-life balance and working conditions were identified as important factors in decision-making regarding emigration and leaving the profession. Subgroup analyses based on gender, type of schooling, fee type and educational background were performed. Only 17.26% of surveyed students were satisfied or very satisfied with the overall prospect of working in the NHS.

**Conclusions** The Ascertaining the career Intentions of UK Medical Students study highlights UK students' views and career intentions, revealing a concerning proportion of those surveyed considering alternative careers or emigration. Addressing factors such as remuneration, work-life balance and working conditions may increase retention of doctors and improve workforce planning efforts.

## STRENGTHS AND LIMITATIONS OF THIS STUDY

⇒ This study represents the largest ever survey of UK medical students, and the largest study investigating medical students' career intentions, providing valuable insights into their future plans.

⇒ This comprehensive survey addresses a topical and critical issue, providing important findings with significant implications for the National Health Service (NHS).

⇒ Due to the cross-sectional design of the study, it captures a 'snapshot' in time, and is thus unable to reflect changes in students' career intentions over time.

⇒ A high consent rate of 71.29% for follow-up studies allows for the possibility of longitudinal validation and observation of changes over time.

⇒ Despite being the largest study of UK medical students, approximately 25.50% of the eligible UK medical students participated, which may introduce selection bias, as it may be that the survey appealed to those already intending to leave the NHS or who were interested in this topic; moreover, a comparison of the survey sample with contemporary demographic data was not possible, as the most recent available data on medical students dated back to 2018.

## INTRODUCTION

Training doctors is a costly investment, and measuring the extent of attrition from the health service in the country of training is crucial to ensure optimal value. Understanding medical students' career plans and trajectories postgraduation is an important factor in effective workforce planning and retention.

There are several factors behind doctors' motivations to emigrate to practise medicine abroad or leave the profession entirely. Commonly cited themes among doctors in the UK include pay erosion and low pay compared with alternative destinations, working conditions within the National Health Service (NHS), well-being, work-life balance and better training opportunities abroad.[1 2]

The UK has 3.2 doctors for every 1000 people, ranking 25th among the Organisation for Economic Co-operation and Development (OECD) countries. This figure also represents the lowest number of doctors per capita among European countries in the OECD.[3] The British government has responded to the issue of an insufficient number of doctors by opening new medical schools and expanding the student capacity of existing ones.[4 5] Recently, there have been proposals to double the number of medical school places as a solution to address the shortage of doctors in the NHS.[6] However, without addressing the issue of doctors leaving the NHS, increasing the number of medical students is unlikely to provide a sustainable long-term solution. Recruitment efforts may be ineffective if the retention of doctors is not simultaneously addressed. This highlights the pressing need for a multifaceted approach that considers both recruitment and retention strategies to effectively address the workforce challenges in the NHS.

## Medical education in the UK

In the UK, after medical school, medical graduates enter the Foundation Programme (FP), a 2-year programme consisting of a series of 4-month or 6-month rotations through various specialties and clinical settings. The successful completion of the programme's first year (Foundation Year 1 (FY1)) provides doctors with full registration with the UK's medical regulator, the General Medical Council (GMC). This registration is recognised internationally. In many cases, individuals who leave the NHS after FY1 rather than immediately following graduation may do so because of the opportunities available with the full registration on completing FY1. Completion of the second year of the programme (FY2) allows applicants to apply for specialist training pathways, such as those in psychiatry, neurosurgery and general practice.[7 8]

To the best of our knowledge, this is the largest study of UK medical students to date. This mixed-methods study aimed to investigate current medical students' career intentions after graduation and on completing the FP, and the motivations behind these intentions. Secondary outcomes included determining which demographic factors alter the propensity to pursue different career paths available to a medical graduate, determining which specialties medical students plan to pursue and understanding current views on the prospect of working in the NHS. These data provide important answers to the current workforce challenges within the NHS and could help address some of the concerns of those making up the future of the profession.

## METHODS

### Study design

AIMS (Ascertaining the career Intentions of UK Medical Students) was a national, multi-centre, cross-sectional study of medical students conducted in accordance with its published protocol.[9] The study employed a non-random sampling method to recruit participants from 44 UK medical schools recognised by the GMC.

A novel, self-administered, 71-item questionnaire was developed. The survey was hosted on the Qualtrics survey platform (Provo, Utah, USA), a GDPR-compliant online platform that supports both mobile and desktop devices. Prior to completing the survey, all participants provided informed consent. All participants were asked to complete the first section of the survey (questions 1–11). Subsequent question visibility was dependent on participants' answers to previous questions. The fewest number of items available to any one participant was 30, and the largest was 43. Questions were structured using a combination of Likert scale matrices, multiple-choice options and free-text entry to broaden the capture of sentiment nuance and improve precision in the data. A copy of the questionnaire and the Participant Information Sheet can be found in online supplemental materials.

### Participant recruitment and eligibility

To minimise bias, a network of approximately 200 collaborators was recruited across 42 medical schools prior to the study launch to ensure equitable access to the survey. All medical students in all year groups were eligible to apply, and positions were advertised via medical student societies, social media and internal medical school newsletters. They were responsible for maximising the response numbers within their year group at their medical schools. Collaborators were instructed to use a range of distribution methods, including social media, internal bulletins/newsletters and email communication. This approach aimed to achieve a representative sample and improve the generalisability of our findings.

In order to qualify for collaborative authorship, students were required to achieve a minimum of 35 responses, or 15% of their year group (whichever number was the lowest). The survey was disseminated between 16 January 2023 and 27 March 2023, by the AIMS Collaborative.

To be eligible for participation, individuals must have been actively enrolled in a UK medical school acknowledged by the GMC and listed by the Medical School Council (MSC) (online supplemental materials). Certain new medical schools have received approval from the GMC but were yet to admit their inaugural cohort of students at the time of data collection. As they had no medical students, these schools were therefore excluded from our study.

### Data collection

The survey consisted of five parts. Part 1 involved a background and demographics section, which all participants were required to answer. In Part 2, participants were asked to indicate their intended career paths immediately after graduation and after foundation training (if applicable). Part 3 explored the factors influencing their decision-making. Part 4 surveyed their current specialty preferences. The final part featured a free-entry text box inviting participants to articulate how the prospect

of working in the NHS could be improved. Consent for follow-up studies was also obtained in this section.

## Data processing and storage

Each response was restricted to a single institutional email address to mitigate the risk of data duplication. Any replicated email entries were removed prior to data analysis. In cases where identical entries contained distinct responses, the most recent entry was retained. Entries where respondents did not provide a valid institutional email address were removed prior to data analysis to preserve the integrity of the study.

## Quantitative data analysis

Descriptive analysis was carried out with Microsoft Excel (V.16.71) (Arlington, Virginia, USA), and statistical inference was performed using RStudio (V.4.2.1) (Boston, Massachusetts, USA). Tables and graphs were generated using GraphPad Prism (V.9.5.0) (San Diego, California, USA). ORs, CIs and p values were computed by fitting single-variable logistic regression models to explore the effect of various demographic characteristics on students' career intentions. CIs were calculated at 95% level. We used $p<0.05$ to determine the statistical significance for all tests.

The findings of this study were reported in accordance with the Strengthening the Reporting of Observational Studies in Epidemiology guidelines.[10]

## Planned subsequent analyses

The comprehensive scope of the AIMS questionnaire requires separate analyses for different components. Future works will specifically focus on the data obtained in parts 4 (specialty preference) and 5 (qualitative responses) of the survey. This approach ensures robust evaluation of these data and their implications, with a full thematic analysis planned for the qualitative data collected.

## Patients and public involvement

In the preparatory phase of the study, an informal focus group convened, comprised medical students at various training stages. These students contributed insights on potential negative aspects of the medical profession within the UK, posited as potential influences on decisions to pause or leave medical training in the UK. In addition, advice was sought from senior clinicians on this topic, providing a more balanced understanding of the issues at hand.

## RESULTS
## Demographics

In total, 10 486 students across all 44 medical schools in the UK participated in the survey (online supplemental figure 1). This represents approximately 25.50% of the medical student population in the UK (n=41 860), according to the latest accessible GMC report on medical student numbers.[11] The mean response number per

**Table 1** Demographic characteristics of participants

| Characteristic | Number (%) |
|---|---|
| Ethnicity | |
| White | 5838 (55.67) |
| Asian or Asian British | 3027 (28.87) |
| Black, Black British, Caribbean or African | 529 (5.04) |
| Mixed or multiple ethnic groups | 555 (5.29) |
| Other | 410 (3.91) |
| Prefer not to say | 127 (1.21) |
| Gender | |
| Female | 6977 (66.54) |
| Male | 3429 (32.70) |
| Non-binary | 64 (0.61) |
| Prefer not to say | 16 (0.15) |
| Level of education | |
| Postgraduate | 1873 (17.86) |
| Undergraduate | 8613 (82.14) |
| Previous schooling | |
| Private education | 3605 (34.38) |
| State education | 6609 (63.03) |
| Prefer not to say | 272 (2.59) |
| Fee status | |
| Home | 9207 (87.80) |
| European Union (EU) | 419 (4.00) |
| International (non-EU) | 860 (8.20) |
| Current year of study | |
| Year 1 | 1963 (18.72) |
| Year 2 | 2152 (20.52) |
| Year 3 | 1952 (18.62) |
| Year 4 (not penultimate year) | 947 (9.03) |
| Penultimate year | 1989 (18.97) |
| Final year | 1483 (14.14) |
| Age | |
| Median (range) | 22 (17–48) |
| Total | 10 486 (100.00) |

medical school was 244, and the median was 203 (IQR 135–281). A breakdown of the response numbers per medical school can be found in the Supplemental Materials. The median age for participants was 22 (IQR 20–23). Although responses were obtained from all year groups, there were relatively fewer responses from students in the 'Year 4 (not penultimate year)' category, likely due to a smaller number of students in intercalating courses or schools with 6-year medical programmes, rather than the conventional 5-year curriculum. Among the participants, 66.5% were female (n=6977), 32.7% were male (n=3429), 0.6% were non-binary (n=64) and 16 individuals preferred not to disclose their gender (table 1).

## Career intentions

All participants were asked their current career intention for immediately after graduation, as shown in online supplemental table 1. The majority of participating students (8806/10 486, 83.98% (CI 83.26%, 84.67%)) planned to complete both years of the UK's foundation training, FY1 and FY2; 10.50% (CI 9.93%, 11.10%) intended to complete FY1 and then emigrate to practise medicine (n=1101); 1.26% (CI 1.06%, 1.49%) planned to complete FY1 and then permanently leave the profession (n=132); 0.99% (CI 0.82%, 1.20%) intended to leave medicine permanently immediately after graduation (n=104); 2.10% (CI 1.84%, 2.39%) planned to emigrate to practise medicine abroad immediately after graduation (n=220) and 1.17% (CI 0.98%, 1.40%) intended to take a break or undertake further study postgraduation (n=123).

Participants intending to complete both years of the FP were then asked their intentions thereafter; the results can be seen in online supplemental table 2. Of these 8806 respondents, 48.76% (n=4294, CI 47.72%, 49.81%) planned to enter specialty training in the UK immediately after the FP; 21.11% (n=1859, CI 20.27%, 21.98%) intended to enter a non-training clinical job in the UK (a common form of 'F3' year, including posts such as junior clinical fellowship or clinical teaching fellowship, or working as a locum doctor). These positions, while clinical in nature and valuable for gaining practical experience, do not typically contribute to full accreditation within a medical specialty, and are thus termed 'non-training'. A further 23.52% of participating students (n=2071, CI 22.64%, 24.42%) intended to emigrate to practise medicine abroad, while 5.85% (n=515, CI 5.38%, 6.36%) planned to take a break or undertake further study. Sixty-seven of the participating students (0.76%, CI 0.60%, 0.97%) planned to leave medicine permanently after FY2.

A total of 32.35% of the surveyed medical students (n=3392/10 486, CI 31.46%, 33.25%) intended to emigrate to practise medicine, either immediately after graduation (n=220/3292, 6.49%, CI 5.71%, 7.36%), after completion of FY1 (n=1101/3292 32.46%, CI 30.90%, 34.05%) or after FY2 (n=2071/3292, 61.06%, CI 59.40%, 62.68%). These students were asked their likelihood of their return to UK medicine (return prospects): 49.56% (n=1681, CI 47.88%, 51.24%) planned to return after a few years, while 7.87% (n=267, CI 7.01%, 8.83%) intended to return after completion of their medical training abroad. The remaining 42.57% (n=1444, CI 40.92%, 44.24%) of those participating students planning on emigrating indicated no intentions to return (online supplemental figure 2A). Of those favouring emigration immediately after graduation, 80.91% did not intend to return to the UK (n=178/220, CI 75.20%, 85.55%). This number decreased to 60.03% (n=661/1101, CI 57.11%, 62.89%) in those planning to emigrate after completing FY1 and 29.21% (n=605/2071, CI 27.29%, 31.21%) in those planning to emigrate after completing FY2, as demonstrated in online supplemental figure 2B.

All participating students intending to emigrate to practise medicine were asked the countries to which they were considering emigrating via a free-entry text box. Students were able to list multiple locations or express if they were undecided. A total of 4115 responses were received from 3392 students. 25.03% (n=849) did not express a preference for any particular destination (figure 1). The remaining 2543 medical students listed 3266 destination preferences. Australia was the most commonly mentioned destination (42.35%), followed by New Zealand (18.03%), the USA (10.38%) and Canada (10.29%).

A total of 303/10 486 (2.89%, CI 2.59%, 3.23%) of surveyed medical students planned to leave the profession entirely, either immediately after graduating (n=104/303, 34.32%, CI 29.20%, 39.84%), after completion of FY1 (n=132/303, 43.56%, CI 38.1%, 49.19%) or after completion of FY2 (n=67/303, 22.11%, CI 17.8%, 27.12%). Students intending to leave the profession were asked the alternative industries they were considering for their future careers (figure 1). 21.12% (n=64/303) of those planning to leave the profession did not yet have an industry in mind. Of the remaining 78.88%, career destinations mentioned most often included consulting, technology, financial services and law.

## Career intention subanalyses

Subanalysis of career intentions after graduation by year of study revealed an overall increase in the proportion of surveyed students intending to complete the FP as they progressed in their medical studies (online supplemental figure 3). Online supplemental figures 3 and 4 highlight the surveyed students' career intentions after graduation and FP, respectively, by year group.

Subanalysis of career intentions after completion of FY2 by current year of study revealed a significant decrease in the proportion of surveyed students looking to enter specialty training as they progressed in their medical studies (online supplemental table 4). By contrast, intentions to emigrate, permanently leave the profession and assume non-training clinical positions also increased as students advanced through medical school (figure 2).

Subanalysis of the subgroup intending to leave medicine (n=303, 2.89%) revealed a significant difference in the proportion of surveyed students taking this decision by various demographic characteristics, as highlighted in table 2. Specifically, males were significantly more likely to plan to leave medicine than females (OR 2.61, CI 2.08, 3.30, p<0.00001), and state-educated students had a higher likelihood of planning to leave medicine compared with privately educated students (OR 1.28, CI 1.01, 1.62, p=0.04). However, no statistically significant difference between home students and non-home students, including international and European Union (EU) students, was identified (OR 1.26, CI 0.71, 2.06, p=0.39). Similarly, we did not find a statistically significant

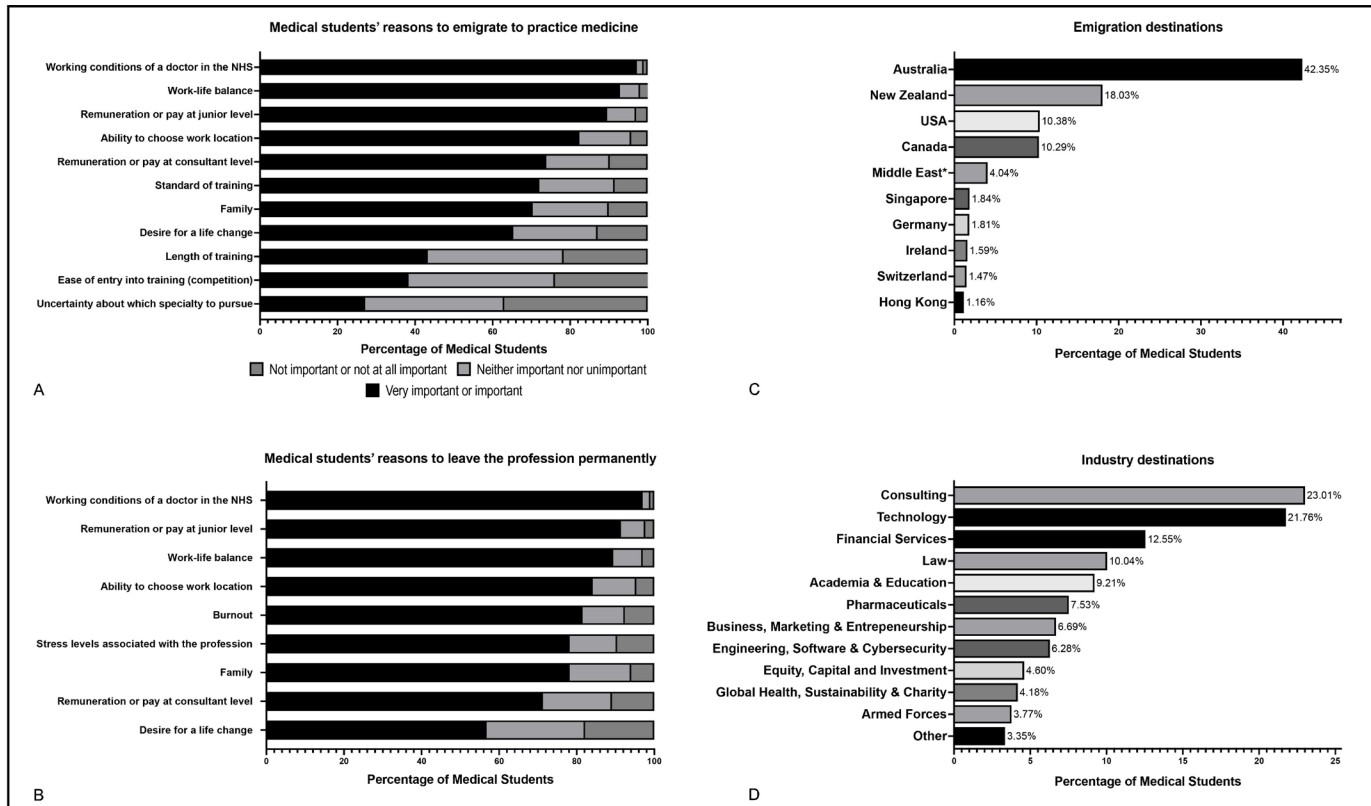

**Figure 1** (A) Importance of factors influencing medical students' intention to emigrate and practise medicine; (B) importance of factors influencing medical students' intention to leave the medical profession entirely and seek an alternative career; (C) locations cited as potential destinations by students who intend to emigrate to practise medicine; (D) preferred industries to work in by those intending to leave medicine. *Several respondents cited the Middle East or Gulf region rather than specifying which country; these responses were grouped with individual destinations in the region.

difference between undergraduates and postgraduates in their likelihood of planning to leave medicine (OR 1.29, CI 0.94, 1.80, p=0.124).

We subanalysed the group of surveyed students intending to emigrate to practise by ethnicity, gender, stage of training, educational background and previous schooling (table 2). Males were significantly more likely to plan to emigrate to practise medicine than females (OR 1.17, CI 1.07, 1.27, p<0.001). Postgraduate students were significantly more likely to plan to emigrate to practise medicine than undergraduate students (OR 1.20, CI 1.08, 1.33, p<0.001). Privately educated students were significantly more likely to plan to emigrate to practise medicine than their state educated peers (OR 1.26, CI 1.15, 1.37, p<0.00001). Non-home students (international and non-EU fees) were considerably more likely to plan to emigrate to practise medicine than home students (OR 2.33, CI 1.92, 2.84, p<0.00001).

We also performed demographic subanalysis on participating students' likelihood to return to the UK if emigrating abroad (online supplemental table 5). Males were significantly less likely to plan to return to the UK after emigrating to practise medicine than females (OR 0.65, CI 0.56, 0.75, p<0.00001). Postgraduates were less likely to plan to return to the UK after emigrating to practise medicine than undergraduates (OR 0.85, CI

0.71, 1.00, p=0.05). Privately educated students were significantly less likely to plan to return to the UK after emigrating to practise medicine than state educated students (OR 0.77, CI 0.67, 0.89, p<0.001). Non-home students (international and EU fees) were significantly less likely to plan to return to the UK after emigrating to practise medicine than home students (OR 0.18, CI 0.14, 0.23, p<0.00001).

## Reasons for students' decisions and overall view of aspects of working in the NHS

Once surveyed students had indicated their intended career option, they were asked the importance behind each of the factors below in their decision to do so. A series of Likert scale matrices were used, with options varying from 'very important' to 'not at all important'. The elements used in the matrices were compiled by the authors through a review of academic and grey literature, social media and input from other clinicians. Students' reasons for planning to leave the NHS, either by emigrating or leaving the profession entirely, can be found in figure 1A,B. For those not entering either the FP or specialty training immediately after completion of medical school or foundation training, burnout and the ability to choose their working location were the most

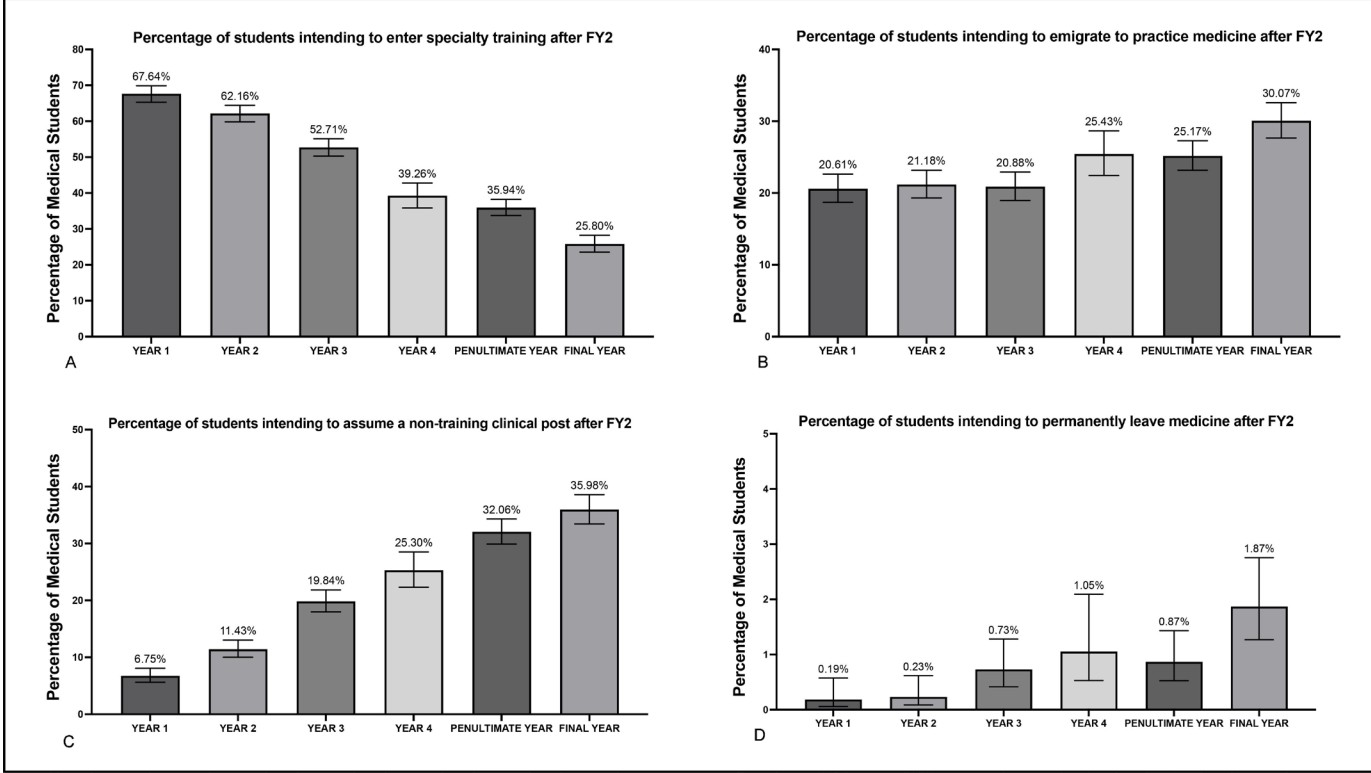

**Figure 2** Proportions of students by year of study (with 95% CIs) intending to (A) directly enter specialty training after Foundation Year 2 (FY2); (B) emigrate to practise medicine after FY2; (C) enter a non-training clinical post after FY2, for example, as a locum doctor or clinical fellow; (D) leave medicine permanently after FY2 to pursue an alternative career. 'Year 4' represents students in their fourth year of study, but not their penultimate year. Percentages in figures reflect the proportion of students in each year group for each intention.

important factors in this decision. The full results can be found in online supplemental figures 4 and 5.

Remuneration at junior level, work-life balance, autonomy over choice of location and the working conditions of doctors in the NHS were identified as the most important factors for surveyed students intending to emigrate to practise medicine (figure 2A). This was also the case for those planning to leave medicine, with the addition of nearly 82% of surveyed students listing burnout as an important or very important reason to abandon the profession (figure 2B).

To better ascertain the surveyed student population's overview of working in the NHS, participants were asked to share their degree of satisfaction with several aspects of working in the NHS. Likert scale matrices were again used in a similar fashion, with options ranging from 'very satisfied' to 'not at all satisfied'. Figure 3 illustrates these findings. Less than 6% of the surveyed medical student population reported feeling satisfied or very satisfied with remuneration at junior level, work-life balance, working conditions of a doctor in the NHS and costs associated with training (such as fees for professional/regulatory body memberships and examinations). A sizeable proportion of participants responded with a neutral rating, neither satisfied nor unsatisfied, when asked about certain aspects of their prospective medical training. Specifically, these aspects included pension tax rules as a consultant,

theatre time during the FP, and exposure to their desired specialty during the FP. In cases where participants may not have held strong opinions on a particular aspect, they tended to select the neutral option. Notably, however, only 17.26% of surveyed students were satisfied or very satisfied with the overall prospect of working in the NHS.

## DISCUSSION
### Principal findings
Our findings demonstrate that a high proportion of the surveyed medical students intend to either leave the profession or permanently emigrate to practise medicine. To the best of our knowledge, there are no previous studies to which to compare these results, so it is difficult to gauge how these figures may have changed over time. We have observed that with each successive year of medical school, the students in our sample became less inclined to enter specialty training in the UK without a break, or at all. Specifically, less than a quarter of final-year medical students surveyed intended to enter specialty training immediately after the FP. In total, 35.23% of the surveyed medical students plan to leave the NHS within 2 years of graduating, either to practise abroad or to pursue other careers. Approximately 60% of the surveyed sample of UK medical students was either not satisfied or not at all satisfied with the prospect of working in the NHS.

**Table 2** Demographic subanalysis of students intending to leave the medical profession and of students intending to emigrate to practise medicine

| Demographic subgroup | Number intending to leave medicine (%) | Number intending to emigrate (%) |
|---|---|---|
| Ethnicity | | |
| White | 147 (2.52) | 1938 (33.20) |
| Asian or Asian British | 99 (3.27) | 911 (30.10) |
| Black, Black British, Caribbean or African | 15 (2.84) | 176 (33.27) |
| Mixed or multiple ethnic groups | 24 (4.32) | 191 (34.41) |
| Other | 10 (2.44) | 141 (34.39) |
| Prefer not to say | 8 (6.30) | 35 (27.56) |
| Gender | | |
| Female | 134 (1.92) | 2183 (31.29) |
| Male | 167 (4.87) | 1191 (34.73) |
| Non-binary | 1 (1.56) | 12 (18.75) |
| Prefer not to say | 1 (6.25) | 6 (37.50) |
| Level of education | | |
| Postgraduate | 44 (2.35) | 669 (35.72) |
| Undergraduate | 259 (3.01) | 2723 (31.62) |
| Previous schooling | | |
| Private education | 118 (3.27) | 1287 (35.70) |
| State education | 170 (2.57) | 2024 (30.62) |
| Prefer not to say | 15 (5.51) | 81 (29.78) |
| Fee status | | |
| Home | 276 (3.00) | 2774 (30.13) |
| European Union (EU) | 15 (3.58) | 217 (51.79) |
| International (non-EU) | 12 (1.40) | 401 (46.63) |
| Current year of study | | |
| Year 1 | 21 (1.07) | 645 (32.86) |
| Year 2 | 42 (1.95) | 713 (33.13) |
| Year 3 | 53 (2.72) | 596 (30.53) |
| Year 4 (not penultimate year) | 46 (4.86) | 326 (34.42) |
| Penultimate year | 75 (3.77) | 616 (30.97) |
| Final year | 66 (4.45) | 396 (33.45) |
| Total | 303 (100.00) | 3392 (100.00) |

## Implications

The NHS is facing a critical workforce shortage, with approximately 10 000 doctors relinquishing their licence to practise in 2021, representing a loss of nearly one-tenth of the doctor workforce.[5 12] A British Medical Association survey of 8000 senior doctors determined that 44% of NHS consultants in England plan to leave or take a break from working in the NHS over the next year.[13] Similarly, a recent survey of 4553 junior doctors in the NHS reported that 4 in 10 plan to leave the NHS, with 33% of these wanting to emigrate to another country to work.[14] The combination of these previous surveys of the doctor workforce, and the results of our medical student survey suggest this trend is presently unlikely to improve. The GMC has recognised the problem and called for immediate action to mitigate the exodus of doctors from the NHS to more attractive employers.[15]

Countries within the anglosphere, namely Australia, New Zealand, the USA and Canada, were the most widely cited destinations for students intending to emigrate. This is perhaps unsurprising given the higher salaries, reports of improved work-life balance, and the fact that these countries' primary language is English.[16] Our study's findings align with previous literature highlighting doctors' leading reasons for emigration, namely pay, working conditions and work-life balance.[1 17]

This study highlights that a disconcerting proportion of participating students, 32.35% (CI 31.46%, 33.25%), intend to emigrate to practise medicine, with nearly half of these students intending not to return. This

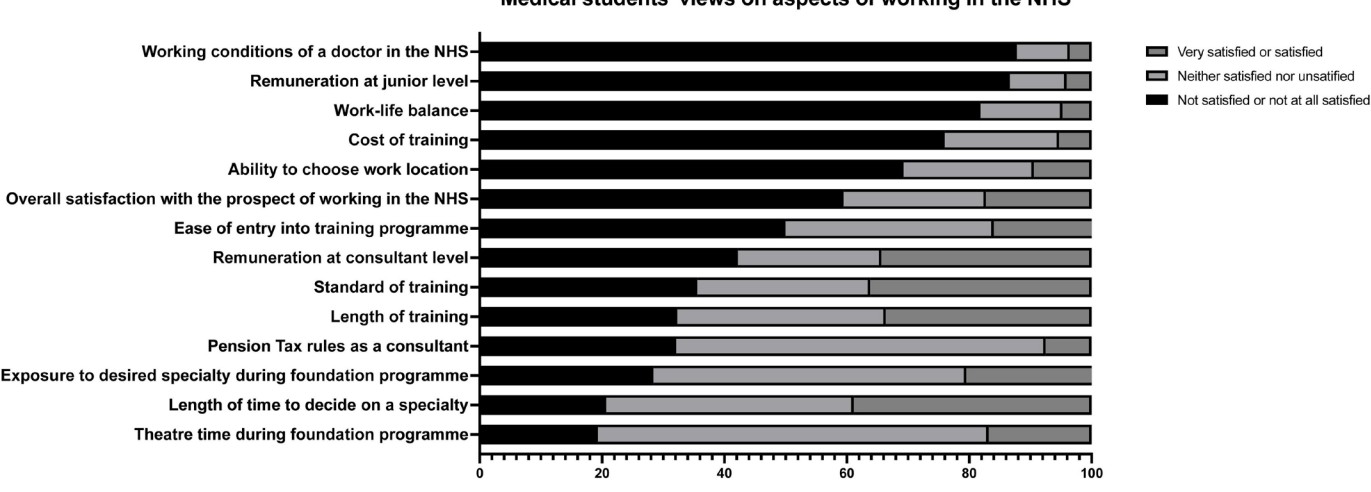

**Figure 3** Medical students' satisfaction levels regarding aspects of working as a doctor in the National Health Service (NHS).

represents a large proportion of the current cohort of medical students. Despite these figures, there remains great uncertainty in this area. It is important to note that a considerable number of students who initially express an intention to emigrate temporarily may ultimately choose to stay abroad permanently.[17] Similarly, some students who do not intend to return to the UK may change their minds in the future. Students paying EU or international fees reported significantly higher intentions to emigrate permanently. The stage at which students intend to emigrate appears to be related to the likelihood of return. Importantly, our study suggests that the proportion of students who intend to leave the NHS may be underestimated, as more students express a desire to leave as they progress through medical school. Moreover, once students enter the FP, a proportion may decide to leave the NHS, even if they had not previously intended to do so.

Insights into the emigration intentions of medical students in other nations indicate that a substantial proportion express a desire to emigrate and practise medicine in countries such as the USA and Canada, as well as to the UK. For instance, in one study, it was found that 49.7% of Malagasy medical students and 25.2% of Tanzanian medical students expressed their intention to emigrate to practise.[18] Similarly, in another study, it was revealed that 44.6% of Ugandan medical students planned to emigrate.[19] It is interesting that the observed trends in these low-income and middle-income countries align with those in the UK, despite the latter's significantly larger economy.

Our results indicate that 2.89% of the medical students participating in our study expressed intentions to quit medicine. A study conducted in Kazakhstan identified a similar trend, with 4% of the participants expressing a desire to leave the medical profession altogether.[20] Additionally, again similar to our results, the study reported a pattern in which medical students in junior years were

less inclined to express a desire to leave the profession compared with students in senior years.[20]

In addition to the 35.24% of sampled medical students intending to quit the NHS within 2 years of graduating, a considerable proportion of participating students (21.11%, CI 20.27%, 21.98%) intended to assume a non-training clinical position in the UK after completing the FP. Participants reported motivations for working in a non-training clinical post in keeping with existing literature surrounding the 'F3' year, with burnout, the ability to choose work location, travel and a greater earning potential evidently being the most compelling reasons to do so.[21 22] Furthermore, in this aspect, we report an increase in intention to not take up specialty posts immediately after the Foundation Programme, with an increase from 6.75% (CI 5.62%, 8.08%) of first-year students to 35.98% (CI 33.45%, 38.59%) of final year students. A contributing factor to this scenario could be a significant increase in competition ratios for specialty training posts, partly due to increasing medical student places and no corresponding increase in the number of training posts available (eg, neurosurgery ST1 competition ratio was 3.9 in 2013 vs 15.94 in 2022).[23] Without corresponding increases to specialist training posts, increases in medical school places may be ineffective in doctor retention.

Historically, the vast majority of medical graduates pursued specialty training immediately after completing their FP; for instance, in 2010, 83.1% of doctors entered specialty training after completing FY2. However, after steadily decreasing year-on-year, this percentage was only 34.9% of doctors in 2019.[7] The UK Foundation Programme Office has not repeated the survey since then, so surmising how these statistics may have changed in the interim is difficult. Our findings indicate that less than half of the medical students surveyed intended to enter specialty training after the FP, with a negative correlation between medical student seniority and intention to enter specialty training with no break, or at all. Only 25.80% of

participating final-year students intended to do so. In the UKFPO survey, those doctors had experienced the negative aspects of the profession. As such, it is concerning to observe this decline in interest among medical students, who have yet to formally begin their careers in medicine.

The findings of our study also align with existing literature on the factors influencing junior doctors' career decisions. Consistently, previous research emphasises the significance of working conditions, location and earnings in shaping these decisions.[1 2 21 22 24–27] Challenging work environments, long hours and inadequate support contribute to disillusionment, burnout and a desire to pursue alternative career paths.[28] Similarly, the autonomy to choose work location emerges as a key factor in medical students, echoing findings among junior doctors. Earnings have consistently been identified as an influential factor for both junior doctors and medical students.[1 2 16 17 21–27] Financial considerations impact their quality of life, student loan repayments and long-term financial stability. The allure of higher salaries and better earning potential in other healthcare systems or professions can attract medical graduates away from NHS training programmes. Addressing working conditions, providing career advancement opportunities, ensuring internationally competitive salaries, and considering location preferences can improve the ability to attract and retain talented professionals. Our study contributes to the growing body of literature by including medical students and supports the notion that working conditions, location and earnings are significant factors influencing junior doctors' decisions to enter or remain in training. These findings underscore the importance of addressing these factors to create a supportive and appealing environment for junior doctors, ultimately promoting better retention rates within the NHS.

Furthermore, our results suggest that the recent calls for dramatic increases in medical school places are unlikely to resolve the NHS staffing shortages. The MSC responded to the original call to increase places by 5000 students by stating multiple barriers, including cost, clinical placement capacity and the lack of a strategic approach to growth. It is estimated that to increase medical schools' capacity by just 5000 places, approximately £1 billion per annum would be required.[29] Additionally, the training of medical students heavily relies on clinical exposure, which in turn is dependent on availability of clinical teaching staff, facilities for training and opportunities.[6] Without a corresponding increase in clinical placement capacity, an increase in medical student places may lead to a decline in the standard of medical education. Our results indicate that increases in medical student places via expansion of existing medical schools or the establishment of new medical schools may not result in proportionate increases in doctors wishing to remain in the NHS. Any attempts to reverse the NHS workforce challenge may benefit from prioritising doctor retention. In this paper, we have highlighted the reasons driving medical students to plan for careers outside of the NHS; addressing these problems is likely to result in improved retention rates.

While there have been studies that (1) explore which specialties junior doctors or medical students intend on pursuing, and exploring factors attracting them to said specialties[30–52]; (2) focus on reasons why doctors are leaving the UK[1 2 24 53]; (3) explore how medical students and junior doctors feel about specific aspects of working within the NHS[25–27 54] and (4) investigate the desire for a career break post-FY2,[21 22] there have been no recent, high-powered studies explicitly aimed at medical students, irrespective of current career ambitions or seniority, investigating overall career intentions and correlating it with demographic factors and medical student seniority. Any statistically significant differences in career intentions between demographic subgroups should be considered carefully and discussed within the correct context. Further studies are required to fully elucidate the reasons behind these disparities.

### Limitations

When interpreting this study's results, there are important limitations to consider. First, the study's cross-sectional nature means we are unable to gauge how students' career intentions may have changed or will change. To address this, we have asked all participants for consent to participate in an anticipated follow-up study, which will enable validation of responses and measurement of change over time; for this, we obtained a 71.29% consent rate.

While this study represents the largest ever study of UK medical students, it is worth noting that approximately 25.50% of the total population of medical students participated. Consequently, we cannot exclude the possibility of selection bias, both from students not seeing the study invitation and others electing not to participate. It may be that this survey appealed to those already intending to leave the NHS or who were interested in the topic. In the context of the UK's medical student population, females were seemingly over-represented in our study despite concerted efforts to ensure equitable outreach during our study advertising phase (57.05% vs 66.50%, respectively).[11] However, the availability of recent demographic data for comparison is limited, with the most recent available data pertaining to the 2018 cohort of medical students.[11]

Additionally, the questions in our survey instruct students to be definitive even when they might not yet have an idea of their career plans, particularly for those in the younger years of medical school. For purposes of brevity and mitigation of survey fatigue, the survey did not provide exhaustive response options. As a result, some decision-making factors may have been omitted. To address this, a free-entry text box was available for participants to supplement their answers. Finally, it should be emphasised that the respondents were medical students who may have limited knowledge of the realities of working in the NHS. Their current reported perceptions may change once they begin their careers in the NHS.

## Conclusion

This study highlights that an alarming proportion of surveyed medical students intend to leave the profession or emigrate to practise medicine. The proportion of students in our sample who plan to leave the NHS within 2 years of graduating is considerable, representing a potential loss of valuable medical talent. Alarmingly, the majority of participating medical students were either not at all satisfied or not satisfied with the prospect of working in the NHS. Additionally, an increasing proportion of the surveyed students intended to take up non-training clinical positions, which could reduce the availability of highly skilled doctors in the NHS. The findings of this study emphasise the urgency of addressing the factors that are driving the exodus of doctors from the NHS and suggest that increased recruitment of medical students may not provide an adequate solution to staffing challenges. The causes of the problem are complex, and finding a solution will require a multifaceted approach. Steps could include improving work-life balance, increasing salaries, addressing the growing competition for specialty training posts and promoting greater flexibility in career pathways. Undoubtedly, the continued loss of skilled professionals from the NHS represents a significant concern, so it is critical to consider means of reversing this trend.

**Acknowledgements** We would like to thank all students who participated in this study. We would also like to thank Mario K Teo and Crispin C Wigfield for their advice in the earlier stages of the study.

**Collaborators** The AIMS Collaborative: Mario K Teo, Crispin C Wigfield, Dania Al-Hashimi, Maeve K Mulchrone, Alisha Pervaiz, Heather A Lewis, Anson Wong, Buzz Gilks, Charlotte Casteleyn, Sara Kidher, Erin Fitzsimons-West, Tanzil Rujeedawa, Meghna Sreekumar, Eliza Wade, Juel Choppy-Madeleine, Yasemin Durmus, Olivia King, Yu Ning Ooi, Malvi Shah, Tan Jih Yih, Samantha Burley, Basma R Khan, Emma Slack, Rishik S Pilla, Jenny Yang, Vaishvi Dalal, Brennan L Gibson, Emma Westwood, Brandon S H Low, Sara R Sabur, Wentin Chen, Maryam A Malik, Safa Razzaq, Amardeep Sidki, Giulia Cianci, Felicity Greenfield, Sajad Hussain, Alexandra Thomas, Annie Harrison, Hugo Bernie, Luke Dcaccia, Linnuel J Pregil, Olivia Rowe, Ananya Jain, Gregory K Anyaegbunam, Syed Z Jafri, Arthur Handscomb, Sudhanvita Arun, Alfaiya Hashmi, Ankith Pandian, Joseph R Nicholson, Hannah Layton-Joyce, Kouther Mohsin, Matilda Gardener, Eunice C Y Kwan, Emily R Finbow, Sakshi Roy, Zoe M Constantinou, Mackenzie Garlick, Clare L Carney, Samantha Gold, Bilal Qureshi, Daniel Magee, Grace Annetts, Tamara Magallon-McGeorge, Khyatee Shah, Kholood T Munir, Timothy Neill, Gurpreet K Atwal, Anesu Kusosa, Anthony Vijayanathan, Mia Mäntylä, Momina Iqbal, Sara Raja, Tushar Rakhecha, Muhammad H Shah, Pranjil Pokharel, Ashna Anil, Kate Stenning, Katie Appleton, Keerthana Uthayakumar, Rajan Panacer, Yasmin Owadally, Dilaxiha Rajendran, Harsh S Modalavalasa, Marta M Komosa, Morea Turjaka, Sruthi Saravanan, Amelia Dickson, Jack M Read, Georgina Cooper, Wing Chi Do, Chiamaka Anthony-Okeke, Daria M Bageac, David C W Loh, Rida Khan, Ruth Omenyo, Aidan Baker, Imogen Milner, Kavyesh Vivek, Manon Everard, Wajiha Rahman, Denis Chen, Michael E Bryan, Shama Maliha, Vera Onongaya, Amber Dhoot, Catherine L Otoibhi, Harry Donkin-Everton, Mia K Whelan, Claudia S F Hobson, Anthony Haynes, Joshua Bayes-Green, Mariam S Malik, Subanki Srisakthivel, Sophie Kidd, Alan Saji, Govind Dhillon, Muhammed Asif, Riya Patel, Jessica L Marshall, Nain T Raja, Tawfique Rizwan, Aleksandra Dunin-Borkowska, James Brawn, Karthig Thillaivasan, Zainah Sindhoo, Ayeza Akhtar, Emma Hitchcock, Kelly Fletcher, Lok Pong Cheng, Medha Pillaai, Sakshi Garg, Wajahat Khan, Ben Sweeney, Ria Bhatt, Madison Slight, Adan M I Chew, Cameron Thurlow, Kriti Yadav, Niranjan Rajesh, Nathan-Dhruv Mistry, Alyssa Weissman, Juan F E Jaramillo, William Thompson, Gregor W Abercromby, Emily Gaskin, Chloe Milton, Matthew Kokkat, Momina Hussain, Nana A Ohene-Darkoh, Syeda T Islam, Anushruti Yadav, Eve Richings, Samuel Foxcroft, Sukhdev Singh, Vivek Sivadev, Guilherme Movio, Ellena Leigh, Harriet Charlton, James A Cairn, Julia Shaaban, Leah Njenje, Mark J Bishop, Humairaa Ismail, Sarah L Henderson, Daniel C Chalk, Daniel J Mckenna, Fizah Hasan, Kanishka Saxena, Iona E Gibson and Saad Dosani.

**Contributors** All the authors meet the ICMJE criteria for authorship. TF responsible for conceptualisation. TF responsible for obtaining funding and ethical approval. TF responsible for collaborator recruitment and management. TF responsible for project administration. TF and AMC responsible for designing the survey. TF responsible for writing the manuscript. OF and RJS responsible for statistical quantitative analysis. All authors responsible for editing and revising the manuscript. RH responsible for supervision. TF is the guarantor. All authors have read and approved the manuscript.

**Funding** Queens' College, University of Cambridge. The institution has had no role in the design of the study, nor collection, analysis, and interpretation of data and in writing the manuscript.

**Competing interests** None declared.

**Patient and public involvement** Patients and/or the public were not involved in the design, or conduct, or reporting, or dissemination plans of this research.

**Patient consent for publication** Not applicable.

**Ethics approval** This study involves human participants and ethical approval was granted by the University of Cambridge Research Ethics Committee (reference PRE.2022.124) on 5 January 2023. Participants gave informed consent to participate in the study before taking part.

**Provenance and peer review** Not commissioned; externally peer reviewed.

**Data availability statement** No data are available.

**ORCID iD**
Tomas Ferreira http://orcid.org/0000-0003-1465-522X

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
