## [Reviewer comments · BMJ Open]

ARTICLE DETAILS

TITLE (PROVISIONAL)	Career intentions of medical students in the United Kingdom: a national, cross-sectional study (AIMS Study)
AUTHORS	Ferreira, Tomas; Collins, Alexander; Feng, Oliver; Samworth, Richard; Horvath, Rita; ., the AIMS Collaborative

VERSION 1 – REVIEW

REVIEWER	Lempp, Heidi King's College London, Inflammation Biology
REVIEW RETURNED	15-Jun-2023

GENERAL COMMENTS	This is an important paper to publish in the current climate of low morale and unrelenting pressure in the NHS, post Covid pandemic and decades of underfunding of the NHS and Social Care. I have the following comments to make for considerations by the authors: Minor corrections: 1) the paper has far too many repeated words, which will disengage the reader/reviewer, e.g. 'intention', 'issue/s', 'address'; 'determining' or 'determination'; 'alluded to', 'our findings', etc. pl go through the paper and find other words, there are many resources available. Major corrections: My background is in qualitative research methods, my feedback will therefore focus on some aspects of your qualitative arm of your study, including a few others. 1) the response rate of 21.4% is in my view too low to make any serious conclusions, this disadvantage has been highlighted in the limitations of your study, however you have not shown any data of the non-responders, which is a pity, due to Qualtrics that collects anonymised data. 2) Who had developed the survey, I could not see any information, how many medical students were involved for example. 3) 71 items seems rather a long questionnaire to me, did the author pilot the survey first of all, nothing was mentioned, why was this information omitted? Could this be one of the reasons of the low syrvey response rate? 4) Why was the qualitative data not analysed in a qualitative computer software programme, and why were none of the accounts included in the paper? The authors summarised the qualitative findings, in my view specific accounts would have provided a much richer picture what is actually bothering medical students, not wanting to stay in the porofession (which was mentioned by the authors in the paper). In additions it would have brocken up the very dense text of discriptive statistics, which I found hard to follow.
--

	5) the validation or credibility of the qual data was not described in detail, apart from discussions within the team, there are many strategies available in the literature. The authors mentioned a few times the subjectivity of qualitative research, why, with adequate qualitative data validation strategies, such a bias would be addressed. 6) I did not understand why under the Thematic Key theme 'The NHS and Society', one of the subtheme was 'changes in government leadership, where is the link? 7) what kind of experiences did the authors have analysing the qualitative data? pl state. 9) bullying and harassment was not included as a potential factor that medical students may want to leave the NHS, this surprised me, as rife in the NHS, according to the literature and during undergraduate medical education. Hence my questions above, who developed the survey? 10) Finally, due to the low response rate, in my view some aspects of the Discussion and Conclusion content are overstated, it may have been the first and largest survey, however the results are not generalisable and this is a serious drawback of the study, which was not highlighted enough.
--	--

REVIEWER	Rangchian, Maryam Hamadan University of Medical Sciences, Pharmacy School
REVIEW RETURNED	07-Jul-2023

GENERAL COMMENTS	First of all, I would like to thank the editor for giving me the opportunity for reviewing this article and the authors for conducting this research. My comments are as follows: The whole manuscript:  - You have mentioned Confidence Interval for some ratios; for example, "The majority of students (8,806/10,486, 83.98% (CI: 83.26%, 84.67%)) planned to complete both years of the UK's foundation training." However, you have not mentioned CI for some other ratios, such as following sentence: "Thirty-two per cent of medical students (n=3,392/10,486, 32.35%) intended to emigrate to practise medicine,". Why? A similar question is the case for the reported OR values. What about their CIs? Abstract:  - Introduction: Please mention the study objective in the abstract. If you face word limit, I suggest that substitute the sentence mentioned in the introduction with a sentence that directly address the study objective. - Method and results: I suggest that eliminate the following marked parts from results and instead provide a more complete method section: 10,486 responses were collected from all 44 medical schools in the UK. To the best of our knowledge, this is the largest ever study of UK medical students. The majority of students (8,806/10,486, 83.98%, CI: 83.26%, 84.67%) planned to complete both years of the Foundation Programme (FP) after graduation, with less than half of these students (4,294/8,806, 48.76%, CI: 47.72%, 49.81%) intending to pursue specialty training thereafter. A subanalysis of career intentions after the FP by year of study revealed a significant decrease in students' intentions to enter specialty training as they advanced through medical school. Approximately
--

	a third of students (3,392/10,486, 32.35%), with intended to emigrate to practice medicine, with 42.57% (CI: 40.92%, 44.24%) of those students not intending to return to the UK. 2.89% of students intended to leave medicine altogether. Remuneration, work-life balance, and working conditions were important factors in students' decision-making regarding emigration and leaving the profession. Subgroup analyses based on gender, type of schooling, fee type, and educational background were performed. Qualitative thematic analysis revealed that the most commonly cited issues included improvements to remuneration, flexibility and work-life balance, general working conditions, staffing levels, and greater autonomy in the location of work.  - Move the sentence "To the best of our knowledge, this is the largest ever study of UK medical students." to the "study limitations and strengths" in the main body of the manuscript. - I suggest the revision of the sentence "Approximately a third of students (3,392/10,486, 32.35%), with intended to emigrate to practice medicine, with 42.57% (CI: 40.92%, 44.24%) of those students not intending to return to the UK. 2.89% of students intended to leave medicine altogether." It is a little hard to understand. Conclusion: I think "as these findings have implications for the future of the medical profession in the UK" can be eliminated. Main body of the article: Introduction: You have implicitly addressed the study aim in the last paragraph, but I suggest that mention it more directly and clear. Methods: - Please clearly state the sampling method. Results: - 66.5% were female (n=6,977). Is this in consistency with the gender distribution of the medical students in UK? I mean, can this finding show that in the case of the gender, the studied sample was not representative of the target population? - Tables: Please set table legends above the tables. Implications - Is British Medical Association (BMA) survey, a specific survey? Is the article "a" appropriate for it or the article "the"? - You have provided some explanation on UK context. Please move these sentences to the introduction section, as a subsection (for example "UK context of the medical education"). In addition, it can be useful for international audience to be provided with a brief explanation about "non-training job". - Please provide more explanation about comparison of the results of the present study with previous studies. It can be helpful for readers if you provide a brief explanation on the results of the relevant studies conducted in other countries. For example I suggest the following article: Faizullina K, Kausova G, Kalmataeva Z, Nurbakyt A, Buzdaeva S. Career intentions and dropout causes among medical students in Kazakhstan. Medicina (Kaunas). 2013;49(6):284-90. PMID: 24248009. - In the last paragraph of the introduction, you have written that "This mixed-methods study investigated current medical students' career intentions after graduation and upon completing the
--	---

	Foundation Programme, and the motivations behind these intentions.” I believe that when investigating the motivations behind the participants’ intentions was one of the research objectives, factors affecting these intentions must be discussed more detailed. So, I think the findings of the references 20-42 that, based on the authors admission, have explored the factors attracting medical students’ intention must be addressed with more details and be compared with the findings of the present study. Limitations: - Please explain why you have stated that “In the context of the UK’s medical student population, females were overrepresented in our study.”?
--	---

VERSION 1 – AUTHOR RESPONSE

Reviewer 1 Comments:

C1: *“The paper has far too many repeated words, which will disengage the reader/reviewer, e.g., 'intention', 'issue/s', 'address'; 'determining' or 'determination'; 'alluded to', 'our findings', etc. Please go through the paper and find other words, there are many resources available.”*

Author’s Response: Thank you for bringing this to our attention. We have carefully reviewed the manuscript and made substantial efforts to replace many of these repetitive phrases throughout the text.

C2: *“The response rate of 21.4% is in my view too low to make any serious conclusions, this disadvantage has been highlighted in the limitations of your study, however you have not shown any data of the non-responders, which is a pity, due to Qualtrics that collects anonymised data.”*

Author’s Response: We acknowledge the concern raised regarding the response rate. However, this study represents the largest survey of UK medical students to date. Below is a graphical representation of the participant size of all UK medical student surveys published on PubMed (n=454). Although the response rate is 21.5%, that is because we used the total medical student population as the denominator for this. It is highly improbable that all medical students heard about the survey and therefore it is a reflection of the proportion of medical students surveyed rather than the proportion of students that had agreed to take part from all invited. In order to address the non-responder data, we made several attempts to obtain demographic information from the GMC and MSC. Unfortunately, the latest data they possess is from 2018, which is unlikely to accurately reflect the current medical student population.

C3: *“Who had developed the survey, I could not see any information, how many medical students were involved for example.”*

Author’s Response: The survey was developed by Authors Tomas Ferreira and Alexander M. Collins, as stated in the Authors’ contributions section. Before survey design, we conducted an informal focus

group involving 8 other medical students to gather their insights on factors that may contribute to medical students' inclination to leave the NHS. We also sought input from Dr. Rita Horvath, the supervising author, and several practicing clinicians in the NHS at various stages of training. Due to the informal nature of the focus group, we believe it would be inappropriate to include it in the methodology section.

C4: "71 items seem rather a long questionnaire to me, did the author pilot the survey first of all, nothing was mentioned, why was this information omitted? Could this be one of the reasons of the low survey response rate?"

Author's Response: Although the survey consisted of 71 items, each participant did not view all of them. The maximum number of items visible to any participant was 43, with the fewest being 30. Additionally, it is worth noting that the "items" included matrices for "views" and "reasons," which involved simple tick boxes. The average completion time for the survey was approximately 5 minutes, and there were incentives to participation, including a £300 prize draw. We do not believe that the study received a low response rate, as it remains the largest survey of UK medical students.

C5: "Why was the qualitative data not analysed in a qualitative computer software programme, and why were none of the accounts included in the paper? The authors summarised the qualitative findings, in my view specific accounts would have provided a much richer picture what is actually bothering medical students, not wanting to stay in the profession (which was mentioned by the authors in the paper). In additions it would have broken up the very dense text of descriptive statistics, which I found hard to follow."

Author's Response: We acknowledge the valid point raised. We agree that presenting specific accounts would offer readers a more comprehensive understanding of the themes. However, we felt that presenting these specific accounts would be beyond the focus of this timely paper. We continue to extract and analyse more information from the data which will be part of a future publication. This will allow for a more in-depth exploration of the students' comments without further extending the length of this paper.

C6: "The validation or credibility of the qual data was not described in detail, apart from discussions within the team, there are many strategies available in the literature. The authors mentioned a few times the subjectivity of qualitative research, why, with adequate qualitative data validation strategies, such a bias would be addressed."

Author's Response: Thank you very much for raising this, we have expanded on the validation and credibility strategies employed in our qualitative data analysis. We hope the revised version provides clearer and more detailed information on this aspect.

C7: "I did not understand why under the Thematic Key theme 'The NHS and Society', one of the subthemes was 'changes in government leadership, where is the link?'"

Author's Response: The subtheme 'changes in government leadership' under the Thematic Key theme 'The NHS and Society' reflects the comments made by students expressing a desire for a change in the current Conservative government to the Labour party or expressing criticisms of the current government. We believed this was a diplomatic and accurate representation of students' comments regarding what steps could be taken to improve the prospects of working in the NHS.

C8: "What kind of experiences did the authors have analysing the qualitative data? Please state."

Author's Response: The authors have gained experience in analysing qualitative data through their involvement in previous unpublished research projects. Additionally, during the data collection phase, we conducted extensive reading on the topic to prepare for the analysis of the substantial qualitative data collected.

C9: "Bullying and harassment was not included as a potential factor that medical students may want to leave the NHS, this surprised me, as rife in the NHS, according to the literature and during undergraduate medical education. Hence my questions above, who developed the survey?"

Author's Response: Thank you for raising this. The survey was developed by two medical students, Tomas Ferreira and Alexander M. Collins, with input from the supervising author (Dr. Rita Horvath) and several practicing clinicians in the NHS at various stages of training (from Foundation doctor to Consultant). Although bullying and harassment are indeed present in the NHS, they were not identified as factors influencing medical students' decision to leave in the informal focus group conducted prior to survey design. While bullying and harassment may drive currently practicing junior doctors away from the NHS, we did not believe it was a prominent factor for medical students considering leaving the NHS before experiencing it. Furthermore, 'bullying and harassment' was not a common theme among the responses to the qualitative question. However, there were references to toxic workplace cultures and media portrayal of doctors.

C10: "Finally, due to the low response rate, in my view some aspects of the Discussion and Conclusion content are overstated, it may have been the first and largest survey, however the results are not generalisable, and this is a serious drawback of the study, which was not highlighted enough."

Author's Response: We appreciate the comment, but we would like to highlight that our survey has received the so far largest response rate ever in a similar study in the UK. Again, while the response rate is 21.5%, that is because we used the total medical student population as the denominator for this.

It is highly improbable that all medical students heard about the survey. With regards to generalisability, we have highlighted the need for cautious interpretation throughout the manuscript and abstract. However, we have taken the editorial comments into consideration and revised phrases such as "a high proportion of medical students" to "a high proportion of participating medical students" to ensure clarity in this regard.

Reviewer 2 Comments:

C1: "You have mentioned Confidence Interval for some ratios; for example, "The majority of students (8,806/10,486, 83.98% (CI: 83.26%, 84.67%)) planned to complete both years of the UK's foundation training." However, you have not mentioned CI for some other ratios, such as following sentence: "Thirtytwo per cent of medical students (n=3,392/10,486, 32.35%) intended to emigrate to practise medicine". Why?

A similar question is the case for the reported OR values. What about their CIs?"

Author's response: We thank the reviewer for raising an important question and apologise for the oversight. The missing confidence intervals for statements have been now added, as well as the confidence intervals for the reported OR values.

C2: "Abstract: Introduction: Please mention the study objective in the abstract. If you face word limit, I suggest that substitute the sentence mentioned in the introduction with a sentence that directly address the study objective."

Authors' Response: We appreciate the reviewer's suggestion. To address this concern, we have revised the abstract to include a clear statement of the study objective, ensuring that it is appropriately highlighted.

C3: "Abstract: Method and results: I suggest that eliminate the following marked parts from results and instead provide a more complete method section: 10,486 responses were collected from all 44 medical schools in the UK. To the best of our knowledge, this is the largest ever study of UK medical students. The majority of students (8,806/10,486, 83.98%, CI: 83.26%, 84.67%) planned to complete both years of the Foundation Programme (FP) after graduation, with less than half of these students (4,294/8,806, 48.76%, CI: 47.72%, 49.81%) intending to pursue specialty training thereafter. A subanalysis of career intentions after the FP by year of study revealed a significant decrease in students' intentions to enter specialty training as they

advanced through medical school. Approximately a third of students (3,392/10,486, 32.35%), with intended to emigrate to practice medicine, with 42.57% (CI: 40.92%, 44.24%) of those students not intending to return to the UK. 2.89% of students intended to leave medicine altogether. Remuneration, work-life balance, and working conditions were important factors in students' decision-making regarding emigration and leaving the profession. Subgroup analyses based on gender, type of schooling, fee type, and educational background were performed. Qualitative thematic analysis revealed that the most commonly cited issues included improvements to remuneration, flexibility and work-life balance, general working conditions, staffing levels, and greater autonomy in the location of work."

Authors' Response: We thank you for the suggestion. We have restructured and amended the abstract as per the Editorial comments and hope this is now satisfactory.

C4: "Abstract: Move the sentence "To the best of our knowledge, this is the largest ever study of UK medical students." to the "study limitations and strengths" in the main body of the manuscript.

Authors' Response: We appreciate your suggestion. The sentence has been removed from the abstract and has been appropriately relocated to the 'Strengths and limitations of this study' section in the main body of the manuscript.

C5: "Abstract: I suggest the revision of the sentence "Approximately a third of students (3,392/10,486, 32.35%), with intended to emigrate to practice medicine, with 42.57% (CI: 40.92%, 44.24%) of those students not intending to return to the UK. 2.89% of students intended to leave medicine altogether." It is a little hard to understand."

Authors' Response: We apologise for the confusion caused by that sentence. We have revised it to improve clarity and ensure better understanding of the intended meaning. Thank you for raising this.

C6: "Abstract: Conclusion: I think "as these findings have implications for the future of the medical profession in the UK" can be eliminated."

Authors' Response: We appreciate your suggestion. The sentence has been removed from the conclusion to avoid redundancy and ensure conciseness.

C7: "Introduction: You have implicitly addressed the study aim in the last paragraph, but I suggest that mention it more directly and clear."

Authors' Response: Thank you for your feedback. We have revised the introduction to explicitly state

the study aim, providing a clearer and direct description of the research objective.

C8: "Methods: Please clearly state the sampling method."

Authors' Response: The revised manuscript includes a clear statement of the sampling method in the methods section for better clarity and transparency. Thank you for raising this.

C9: "Results: 66.5% were female (n=6,977). Is this in consistency with the gender distribution of the medical students in UK? I mean, can this finding show that in the case of the gender, the studied sample was not representative of the target population?" & "Please explain why you have stated that "In the context of the UK's medical student population, females were overrepresented in our study"."

Author's response: According to the GMC report "The state of medical education and practice in the UK: 2020 Reference tables – medical students," female students make up 57.1% of medical students in the UK. In our study, 66.54% of participants were female, indicating an overrepresentation of female medical students in our survey (https://www.gmc-uk.org/-/media/documents/gmc-somep-2020-reference-tables-about-medical-students_pdf-84718237.pdf). We have acknowledged the potential selection bias and representation disparities in our survey in the limitations section of the revised manuscript.

C10: "Tables: Please set table legends above the tables."

Author's response: Thank you for bringing this to our attention. We have made the necessary adjustments and positioned the table legends above the tables for better clarity and organisation.

C11: "Is British Medical Association (BMA) survey, a specific survey? Is the article "a" appropriate for it or the article "the"?"

Author's response: The BMA carries out several surveys of their members regularly, and in this case, we are referring to a specific survey conducted by the BMA (as referenced). We have reviewed the language used and made appropriate adjustments to ensure accuracy and clarity.

C12: "You have provided some explanation on UK context. Please move these sentences to the introduction section, as a subsection (for example "UK context of the medical education"). In addition, it can be useful for international audience to be provided with a brief explanation about "non-training job"."

Author's response: Thank you for your suggestion. We agree that providing contextual information on the UK and clarifying terms such as 'non-training job' would enhance the understanding of an international audience. We have incorporated these suggestions by creating a new subsection in the introduction called 'UK context of medical education' and providing a brief explanation of 'nontraining job' to ensure clarity for an international readership.

C13: "Please provide more explanation about comparison of the results of the present study with previous

studies. It can be helpful for readers if you provide a brief explanation on the results of the relevant studies

conducted in other countries. For example, I suggest the following article: Faizullina K, Kausova G, Kalmataeva Z, Nurbakyt A, Buzdaeva S. Career intentions and dropout causes among medical students in

Kazakhstan. *Medicina (Kaunas)*. 2013;49(6):284-90. PMID: 24248009.”

Author's Response: We thank the reviewer for this insightful comment and helpful suggestion. We acknowledge the importance of providing a comparative analysis of our study's results with those of previous studies conducted in other countries. In response to this suggestion, we have included a discussion of international studies with similar objectives in our revised manuscript, one of which is the article suggested by the reviewer.

C14: “In the last paragraph of the introduction, you have written that “This mixed-methods study investigated current medical students' career intentions after graduation and upon completing the Foundation Programme, and the motivations behind these intentions”. I believe that when investigating the motivations behind the participants' intentions was one of the research objectives, factors affecting these intentions must be discussed more detailed. So, I think the findings of the references 20-42 that, based on the authors admission, have explored the factors attracting medical students' intention must be addressed with more details and be compared with the findings of the present study.”

Author's Response: We agree with your point that a more detailed discussion of the factors affecting participants' intentions would be valuable. However, considering the extensive data collected and the potential lengthiness of the paper, we have made the decision to dedicate a subsequent paper entirely to the in-depth analysis of these reasons and conducting subgroup analysis based on gender differences.

For example, we plan to investigate why female medical students may be more likely to state "Worklife balance" as an important reason for emigrating abroad compared to males. We believe this approach will ensure a focused and concise presentation of the findings while maintaining the readers' interest.

Despite this, we have added a paragraph dedicated to discussing reasons for medical students and junior doctors for emigrating or leaving the NHS, using findings from other studies.